# Stimulatory Effects of Extracellular Vesicles Derived from *Leuconostoc holzapfelii* That Exists in Human Scalp on Hair Growth in Human Follicle Dermal Papilla Cells

Yeo Cho Yoon [1], Beom Hee Ahn [1], Jin Woo Min [2], Kyung Real Lee [3], Sang Hoon Park [4] and Hee Cheol Kang [1,2,*]

[1] Human & Microbiome Communicating Laboratory, GFC Co., Ltd., Hwaseong 18471, Korea; yc.yoon@gfcos.co.kr (Y.C.Y.); bh.ahn@gfcos.co.kr (B.H.A.)
[2] Green & Biome Customizing Laboratory, GFC Co., Ltd., Hwaseong 18471, Korea; jw.min@gfcos.co.kr
[3] SKINDA Dermatology Clinic, Sungnam 13595, Korea; teamdoctor78@naver.com
[4] Department of Plastic Surgery, ID Hospital, Seoul 06039, Korea; spark@idhospital.com
[*] Correspondence: michael@gfcos.co.kr

**Abstract:** Human hair follicle dermal papilla cells (HFDPCs) located in hair follicles (HFs) play a pivotal role in hair follicle morphogenesis, hair cycling, and hair growth. Over the past few decades, probiotic bacteria (PB) have been reported to have beneficial effects such as improved skin health, anti-obesity, and immuno-modulation for conditions including atopic dermatitis and inflammatory bowel disease (IBD). PB can secrete 50~150 nm sized extracellular vesicles (EVs) containing microbial DNA, miRNA, proteins, lipids, and cell wall components. These EVs can regulate communication between bacteria or between bacteria and their host. Although numerous biological effects of PB-EVs have been reported, the physiological roles of *Leuconostoc holzapfelii* (hs-Lh), which is isolated from human scalp tissue, and the extracellular vesicles derived from them (hs-LhEVs) are largely unknown. Herein, we investigated the effects of hs-LhEVs on hair growth in HFDPCs. We show that hs-LhEVs increase cell proliferation, migration, and regulate the cell cycle. Furthermore, hs-LhEVs were found to modulate the mRNA expression of hair-growth-related genes in vitro. These data demonstrate that hs-LhEVs can reduce apoptosis by modulating the cell cycle and promote hair growth by regulation via the Wnt/β-catenin signal transduction pathway.

**Keywords:** extracellular vesicles (EVs); probiotic bacteria; human scalp-derived *Leuconostoc holzapfelii* (hs-Lh); apoptosis; hair growth

## 1. Introduction

Human hair not only affects external appearance and protects the head but also maintains head temperature [1]. Hair loss, known as alopecia, is a common medical disorder that affects all genders. Hair loss can occur as a result of several changes such as seasons, fatigue, stress, hormone imbalances, drugs, pollution, and aging [2]. In a modern society that values outward appearance, hair loss can cause mental stress and lower quality of life, and it can become an obstacle to human relationships or social life [3,4]. Hair follicles (HFs), one of the adnexal components of mammalian skin, are derived from interactions between epithelial and mesenchymal compartments during embryonic development [5,6], and they are the sites of hair growth, which is regulated by growth factors [7]. Hair growth occurs in four phases: anagen (a rapid hair proliferation period), catagen (apoptosis-driven regression period), telogen (a relatively inactive period), and exogen [8–11]. Current, hair loss treatments include cosmetic products, surgical operation (such as hair follicle transplantation), and oral or topical medications such as finasteride (a 5α-reductase inhibitor) and minoxidil (which was developed as a hypertension treatment) [12]. Additionally, blue light treatments have been proposed as additional methods for hair loss treatment [13]. However, these two representative drugs have short-term effects that degrade when treatment is

discontinued [14,15] and side-effects including sexual dyfunction [16], itching, erythema, and dryness [17].

Cell proliferation and apoptosis play pivotal roles in the biology of hair growth. Apoptosis occurs via a series of early, mid, and late apoptosis-related genes, such as caspase-3, Bcl-2, and Bax [18], among which the expression ratio of Bcl-2 to Bax is an important determinant of apoptosis [19,20]. Cisplatin, an anti-cancer drug, can induce apoptosis via the mitochondrial signaling cascade [21,22] and cause hair loss during clinical application [23,24]. In mammalian HF growth and cycling, the Wnt/β-catenin pathway is known to have positive effects [7,25]. Activated β-catenin accumulates in the cytoplasm, translocates into the nucleus, and regulates the expression of transcription factors including Lef/Tcf, which are responsible for HF growth [26].

Extracellular vesicles (EVs) are the smallest membrane vesicles [27] and consist of heterogeneous membrane structures containing trans-membrane proteins and proteins, lipids, DNA, and RNA molecules derived from the host cell [28–30]. Most cell types, eukaryotic and prokaryotic, secrete EVs that play essential roles in cellular function via the evolutionary conserved mechanism of the transport of micro RNA, mRNA, or transcription factors (TFs) to nearby cells [31–34]. EVs, which have various sizes (some smaller than 150 nm in diameter [35]) and inner compositions, regulate mechanisms in biogenesis and critical functions through cell-cell communication as cargo transporters. In their biogenesis, EVs released via the endocytotic pathway, are generally categorized into subtypes such as exosomes, microvesicles, and apoptotic bodies.

Many Lactobacillus and Bifidobacterium species are probiotic bacteria [36]. These species have been reported to modulate the mucosal immune system of the host and to reduce inflammatory responses in both rodent colitis models and patients with inflammatory bowel disease (IBD) [37,38]. Common probiotic bacteria are divided both Gram-positive and Gram-negative bacteria; Gram-positive bacteria consist of a cell wall, membrane, and cytoplasm, which is a simple structure compared to that of Gram-negative bacteria, which also have an outer membrane. Among various effects related to probiotic bacteria, it has been reported that the hydrolyzed Lactobacillus plantarum promotes hair growth in vitro and in vivo [39]. These probiotic bacteria are known to also secrete EVs with DNA, protein, and cell wall components, and they are known to play roles as key messengers [40–43]. Generally, probiotic bacterial EVs (PB-EVs) can be isolated from cell-conditioned medium (CM) supernatants or body fluids and have been reported to be related to physiological and pathological processes [29]. Currently, the various biological effects of PB-EVs are being globally investigated. Among them, it has been reported that *L. plantarum*-derived EVs protect against atopic dermatitis induced by Staphylococcus aureus-derived EVs [44]. According to a recent report, dermal papilla cell (DPC) exosomes can induce the anagen stage, delay the catagen stage of HF growth, and stimulate cell proliferation and differentiation in outer root sheath cells [45]. Similarly, PB-EVs can play various roles, but the effect of *L. holzapfelii*-derived extracellular vesicles (hs-LhEVs), which originate from human scalp tissue, has not yet been determined in regard to hair growth effects.

The aim of the present study was to investigate the hair growth effects of hs-LhEVs on the regulatory mechanism in human hair follicle dermal papilla cells (HFDPCs) and anti-apoptotic effects in cisplatin-induced HFDPCs.

## 2. Materials and Methods

### 2.1. Separation and Isolation of Microorganisms

The microorganisms used in this study were autonomously isolated from human skin (*L. plantarum*), infant feces (*B. longum* and *B. animalis*), and yogurt (*L. acidophilus*). *Leuconostoc holzapfelii* was isolated from the scalps of women in their 20s, and the separation method is as follows. We rubbed gauze on the scalp suspensions obtained via the addition of distilled water and shook the mixture; then, 100 μL of each sample were spread on MRS agar plates under aerobic condition at 37 °C. Single colonies were purified by transferring

them to new MRS agar plates. The GFC120 strain was maintained in a de Man, Rogosa, and Sharpe (MRS) broth medium containing 30% glycerol at -70 °C.

### 2.2. 16S rRNA, pheS and atpA Gene Sequence and Phylogenetic Analysis

Genomic DNA of the GFC1203H strain was isolated using the Genomic DNA isolation Kit (Gene all, South Korea) according to the manufacturer's instructions. The 16S rRNA gene was amplified from chromosomal DNA of the GFC1203H strain using the universal bacteria primer sets 27F/1492R [46]. The full sequence (1500 bp) of the 16S rRNA was assembled with SeqMan software version 7.1 (DNASTAR Inc., Madison, WI, USA) and BioEdit [47]. The 16S rRNA gene sequence similarities between GFC1203H and other related Leuconostoc species were obtained from the GenBank database. Multiple alignments of the sequences were carried out using the program CLUSTAL× [48]. The distance matrices for aligned sequences were calculated using the two-parameter method of Kimura [49]. A phylogenetic tree was constructed with the neighbor-joining method [50] and the maximum-parsimony method [51] using the MEGA7 Program [52]. A bootstrap analysis with 1000 replicates was conducted in order to obtain confidence levels for the branches [53].

### 2.3. Preparation of Probiotic Bacteria

Single colonies of probiotic bacteria (PB) belonging to the Lactobacillus and Bifidobacterium genera were inoculated and pre-incubated in an MRS medium (KisanBio, Seoul, Republic of Korea) at 37 °C for 24 h. Next, the cultured PB were rinsed three times with DPBS to remove residual medium and incubated in 10% skim milk (BD, Franklin Lakes, NJ, USA) at 37 °C for 48 h for use in this study.

### 2.4. Isolation of Extracellular Vesicles

Extracellular vesicles were isolated from supernatants of a cultured probiotic bacteria medium including *Leuconostoc holzapfelii* originating from human scalp tissue. They were centrifuged at 3600 rpm for 20 min and then passed through a 0.22 μm syringe filter to remove possible contaminants. EVs were purified after centrifuging the supernatants at $4000 \times g$ for 1 h and $10,000 \times g$ for 1 h before ultra-centrifugation at $100,000 \times g$ for 2 h (Hitachi, Chiyoda-ku, Tokyo, Japan). The EV-rich pellets were re-suspended to a final volume of 1~2 mL with DPBS and frozen for storage at -80 °C in an ultra-low temperature freezer. The conditioned medium of human-scalp-derived *Leuconostoc holzapfelii* (hs-LhCM), which was the negative control, was used in the experiment after filtering with a 0.22 μm syringe filter.

### 2.5. Nanoparticle Tracking Analysis (NTA)

In order to confirm the morphology, including diameter and particle concentration, of the extracellular vesicles isolated by ultracentrifugation, nanoparticle tracking analysis was conducted with a ZetaView TWIN (Particle Metrix, Meerbusch, DE). PB-EVs (including *L. holzapfelii* isolated from the human scalp) suspended in filtered DPBS at 22.93 °C were irradiated with a blue-light laser wavelength ($\lambda$ = 488 nm). The conductivity of samples was evaluated at 14.14 μS/cm, and filter wavelength was measured with backscatter detection. Samples were measured with dilution (dilution factor is 2000) and an average of 11 on the equivalent sample aliquot. Data were analyzed using ZetaView Software (version 8.05).

### 2.6. Cryo-TEM Analysis

Most exosomes were visualized, after negative staining, with transmission electron microscopy (TEM). First, the exosomes were glow-discharged with a thin formvar/carbon film coated with 200 mesh copper EM grids for 2 min with a glow discharger. Then, in a ventilated fume hood, purified exosomes were fixed with 500 μL of 2% paraformaldehyde for 5 min and diluted to a 1/2 concentration exosome suspension solution on the grid before being incubated for 3 min. Exosomes were then immediately stained with a filtered 1.5% uranyl acetate solution on the surface of the EM grid. Then, the UA solution was removed

from the grid using filter paper, and the grid was rinsed with a distilled water to remove excess staining solution. After that, the grid was dried for 10 min at room temperature and observed via TEM at 80 kV.

### 2.7. Cell Culture and Proliferation Assay

The human hair follicle dermal papilla cells (HFDPCs) were purchased from the American Type Culture Collection (Manassas, VA, USA) and incubated in an alpha-minimal essential medium ($\alpha$-MEM, WelGene Inc., Daegu, Republic of Korea) supplemented with 10% fetal bovine serum (FBS, Hyclone Laboratories Inc., Logan, UT, USA) and 1% penicillin streptomycin (Hyclone Laboratories Inc., Logan, UT, USA) at 37 °C in an atmosphere containing 5% $CO_2$. Cell proliferation was measured using the cell counting kit-8 (CCK-8, Dojindo Molecular Technologies Inc., Rockville, MD, USA) colorimetric assay. The cells ($5 \times 10^4$ cells/well) in a conditioned medium containing 10% FBS were seeded in a 96-well plate and incubated for 24 h. Next, the cells were starved for 24 h and incubated in the presence of various concentrations of hs-LhEVs or 10 µg/mL of other PB-EVs for 24 h at 37 °C in an atmosphere containing 5% $CO_2$. Finally, the cell medium was removed and replaced with 100 µL of fresh conditioned medium containing a 10% CCK-8 reagent in each well, and then the cells were incubated for another 2 h. The sample absorbance was measured at 450 nm against a background control (DPBS) using a microplate reader (BioTek Instruments, Inc., Winooski, VT, USA).

### 2.8. Cell Viability Assay

To confirm anti-apoptotic effects, we investigated cell viability with hs-LhEVs in the presence of cisplatin (Sigma-Aldrich Chemical Co., St. Louis, MO, USA), which is known to cause apoptosis. Cell viability was determined using the cell proliferation reagent WST-1 (Dojindo Molecular Technologies Inc., Rockville, MD, USA). First, cells were seeded in a 96-well plate at $1 \times 10^4$ cells/well in 1 mL of a complete conditioned medium supplemented with 10% fetal bovine serum (FBS, Hyclone Laboratories Inc., Logan, UT, USA) and 1% penicillin streptomycin (Hyclone Laboratories Inc., Logan, UT, USA) before being incubated for 24 h at 37 °C in an atmosphere containing 5% $CO_2$. Cells were simultaneously treated with 100 µM of cisplatin; 1, 2.5, 5, and 10 µg/mL dose-dependent *Leuconostoc holzapfelii*-derived extracellular vesicles (hs-LhEVs); and 10 µg/mL of conditioned medium of human-scalp-derived *Leuconostoc holzapfelii* (hs-LhCM). Then, they were incubated for 24 h at 37 °C in 5% $CO_2$. After 24 h of incubation, 500 µL/well WST-1 reagents were added and incubated for 2 h. The cells were thoroughly mixed on a shaker for 30 s, and the absorbance was measured against a background control with a microplate reader (BioTek Instruments, Inc., Winooski, VT, USA) at 450 nm.

### 2.9. Wound Healing Assay

HFDPCs were seeded 24-well plates to $5 \times 10^4$ cells/well, cultured until they reached 90% confluence, and rinsed with DPBS. A 200 µL sterile plastic pipette tip was used to scratch a single wound in the center of the cell monolayer, and the non-adherent cells were removed with DPBS wash. The wounded cells were incubated with either 10 µg/mL of other PB-EVs or dose-dependent concentrations (1, 2.5, 5, and 10 µg/mL) of hs-LhEVs for 6, 12, and 24 h. The lengths of cell migration from the edge of the injured monolayer were measured at each time point, and images were obtained using a phase-contrast microscope (OPTINITY). The wound healing length was analyzed with Image-Pro Plus v 6.0 (Media Cybernetics, Inc., Bethesda, MD, USA), and each experiment was repeated at least three times.

### 2.10. Propidium Iodide (P.I.) Staining and Cell Cycle Analysis

To analyze changing patterns of cell cycles, HFDPCs were seeded in 6-well plates at $1.2 \times 10^5$ cells/well and cultured for 24 h. Next, the cells were treated with various concentrations of hs-LhEVs for 6, 12, and 24 h. Following incubation, the cells were collected

and washed 3 times with ice-cold DPBS the cells fixed with 70% ethanol at -20 °C for 16 h and treated with 1% RNase A at 37 °C for 30 min prior to being stained with propidium iodide (Sigma-Aldrich Chemical Co., St. Louis, MO, USA). The stained cells were analyzed using the Guava easyCyte™ HT flow cytometry systems (Millipore Sigma, Burlington, MA, USA). All samples collected contained at least 10,000 events, and data were analyzed using Guava software (version 3.1.1).

### 2.11. Reverse Transcription Polymerase Chain Reaction (RT-PCR)

To investigate changing mRNA expression patterns of apoptosis and hair-growth-related genes following treatment with hs-LhEVs, reverse transcription-polymerase chain reaction (RT-PCR) analysis was performed. A TRIzol reagent (Sigma-Aldrich Chemical Co., St. Louis, MO, USA) was used to extract mRNA from hs-LhCM- and hs-LhEV-treated HFDPCs. The spectrophotometric determination of mRNA purity and integrity was performed using a NanoDrop™ 2000/2000c Spectrophotometer (Thermo Fisher scientific, Waltham, MA, USA) and analyzed at 260/280 nm. RT-PCR was conducted with primers for the Bcl-2, Bax, caspase-3, Wnt5A, Wnt10B, β-catenin, activin membrane-bound inhibitor homolog (BAMBI), bone morphogenetic protein 2 (BMP2), lymphoid enhancer-binding factor 1 (Lef1), and versican (VSC) genes, including GAPDH as a house-keeping gene (Table 1). Two micrograms of RNA template were reverse-transcribed using the amfiRivert cDNA synthesis platinum master mix (GenDEPOT, Katy, TX, USA) and amplified by PCR using a C1000 Touch™ thermal cycler (Bio-Rad, Hercules, CA, USA). The PCR program was an initial denaturation at 95 °C for 2 min followed by 35 cycles of 30 s at 95 °C, 60 s at 65 °C, and 5 min at 72 °C. In this study, to evaluate the activities of only hs-LhEV, hs-LhCM was used as a negative control. All results were normalized to a human GAPDH control housekeeping gene, and melting curve analysis was used to confirm the presence of a single PCR product. The PCR amplification was confirmed by electrophoresis using 1% agarose gel. In this study, all mRNA expression experiments were repeated more than three times.

**Table 1.** Sequences of the primers used in this study.

| Gene | Direction | Primer Sequences (5′−3′) | Length of Expected Fragment (bp) |
|---|---|---|---|
| Wnt5A | Sence<br>Anti-sence | 5′-TCCACCTTCCTCTTCACACTGA-3′<br>5′-CGTGGCCAGCATCACATC-3′ | 65 |
| Wnt10B | Sence<br>Anti-sence | 5′-CTTTTCAGCCCTTTGCTCTGAT-3′<br>5′-CCCCTAAAGCTGTTTCCAGGTA-3′ | 87 |
| BAMBI | Sence<br>Anti-sence | 5′-CTCCCGTTTGCACTACAGCTT-3′<br>5′-CTTTGCAACCTGCCCCTTT-3′ | 58 |
| BMP2 | Sence<br>Anti-sence | 5′-GAGGTCCTGAGCGAGTTCGA-3′<br>5′-TCTCTGTTTCAGGCCGAACA-3′ | 57 |
| LEF1 | Sence<br>Anti-sence | 5′-CCCGATGACGGAAAGCAT-3′<br>5′-TCGAGTAGGAGGGTCCCTTGT-3′ | 58 |
| VCAN | Sence<br>Anti-sence | 5′-GGCAATCTATTTACCAGGACCTGAT-3′<br>5′-TGGCACACAGGTGCATACGT-3′ | 103 |
| β-catenin | Sence<br>Anti-sence | 5′-CTGCTGTTTTGTTCCGAATGTC-3′<br>5′-CCATTGGCTCTGTTCTGAAGAGA-3′ | 99 |
| BCL-2 | Sence<br>Anti-sence | 5′-TCCCTCGCTGCACAAATACTC-3′<br>5′-ACGACCCGATGGCCATAGA-3′ | 72 |
| BAX | Sence<br>Anti-sence | 5′-CTGCAGAGGATGATTGCCG-3′<br>5′-TGCCACTCGGAAAAAGACCT-3′ | 63 |
| Caspase-3 | Sence<br>Anti-sence | 5′-TGGTTCATCCAGTCGCTTTG-3′<br>5′-CATTCTGTTGCCACCTTTCG-3′ | 101 |
| GAPDH | Sence<br>Anti-sence | 5′-TGGAAATCCCATCACCATCTTC-3′<br>5′-CGCCCCACTTGATTTTGG-3′ | 56 |

### 2.12. Statistical Analysis

In this study, significance was confirmed using SPSS version 20.0 program (SPSS Inc., Armonk, NY, USA). The paired t-test was used to compare the values measured before and after the test, and the significance was confirmed at the level of $p < 0.05$, $p < 0.01$, $p < 0.001$ as the mean $\pm$ SD of at least three independent experiments.

## 3. Results

### 3.1. 16S rRNA, pheS and atpA Gene Sequence and Phylogenetic Analysis

The comparative analysis of 16S rRNA gene sequences showed that the GFC1203H strain belongs to the Leuconostoc genus and is most closely related to *Leuconostoc holzapfelii* BFE7000T (99.2% similarity), *Leuconostoc citreum* ATCC49370T (98.47%), *Leuconostoc miyukkimchii* M2T (98.04%), *Leuconostoc palmae* TMW2.964T (97.56%), and *Leuconostoc latics* JCM6123T (97.56%) (Figure 1).

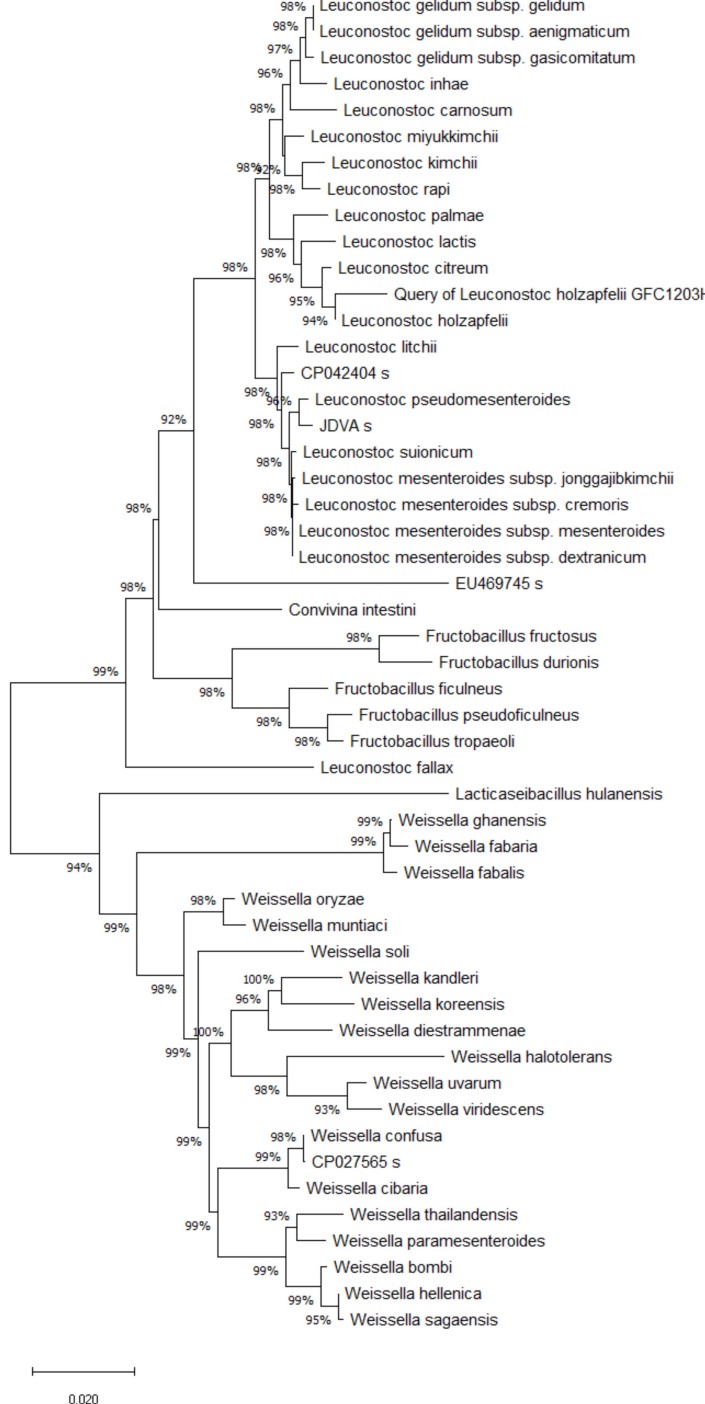

**Figure 1.** Phylogenetic tree based on 16S rRNA gene sequences of *Leuconostoc holzapfelii* isolated from the human scalp.

### 3.2. Probiotic Bacteria Voluntarily Produce Extracellular Vesicles

We isolated EVs from the supernatant of cultured PB including *Bifidobacterium longum* (Bl), *B. animalis* (Ba), *Lactobacillus acidophilus* (La), human-scalp-derived-*Leuconostoc holzapfelii* (hs-Lh), and *Lactobacillus plantarum* (Lp) using ultracentrifugation. The particle size and distribution of isolated PB-EVs were determined with nanoparticle tracking analysis (NTA) (Figure 2A). NTA analysis revealed that the UC-purified PB-EVs had mean diameters of 126.5 nm (BlEVs; mode diameter: 131.4 nm), 127.8 nm (BaEVs; mode diameter: 116.6 nm), 127.7 nm (LaEVs; mode diameter: 127.2 nm), 104.6 nm (hs-LhEVs; mode diameter: 89.1 nm), and 126.7 nm (LpEVs; mode diameter: 116.0 nm) (Figure 2B). The number of particles per mL of EVs and mg of protein concentration isolated from Bl, Ba, La, hs-Lh, and Lp were $2.2 \times 10^{11}$ (per mg: $1.04 \times 10^{10}$), $1.9 \times 10^{11}$ (per mg: $1.12 \times 10^{10}$), $3.8 \times 10^{11}$ (per mg: $1.65 \times 10^{10}$), $2.03 \times 10^{11}$ (per mg: $1.03 \times 10^{10}$), and $4.6 \times 10^{11}$ (per mg: $1.74 \times 10^{10}$), respectively (Figure 2C). For the morphological analyses of purified hs-LhEVs, EVs were measured using Cryo-TEM with negative staining, and the results showed that they had a spherical membrane structure and a lipid bilayer (Figure 2D). These results showed that multiple PB produce EVs of different sizes and that hs-LhEVs are remarkably smaller than other PB-EVs.

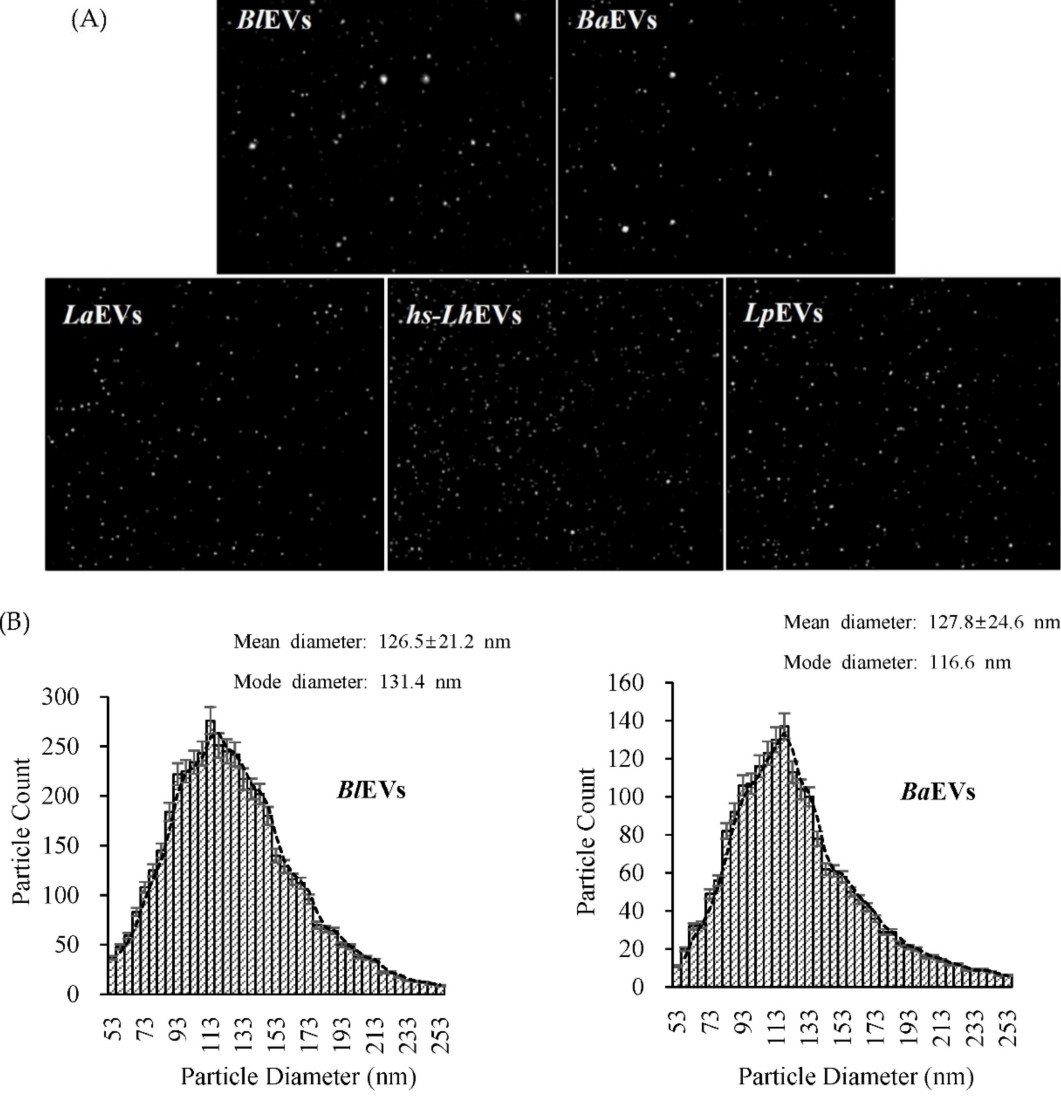

**Figure 2.** *Cont.*

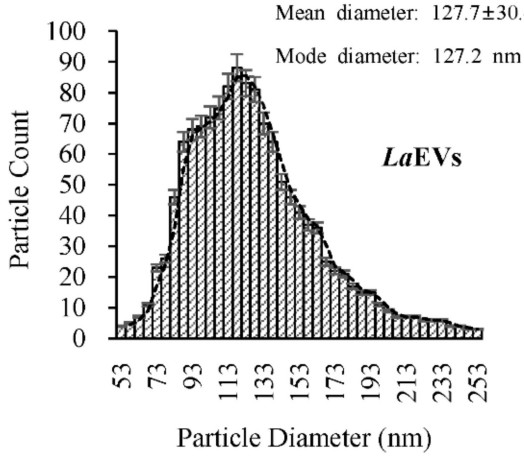

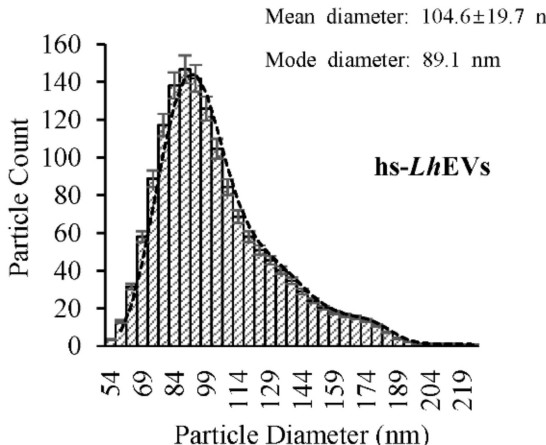

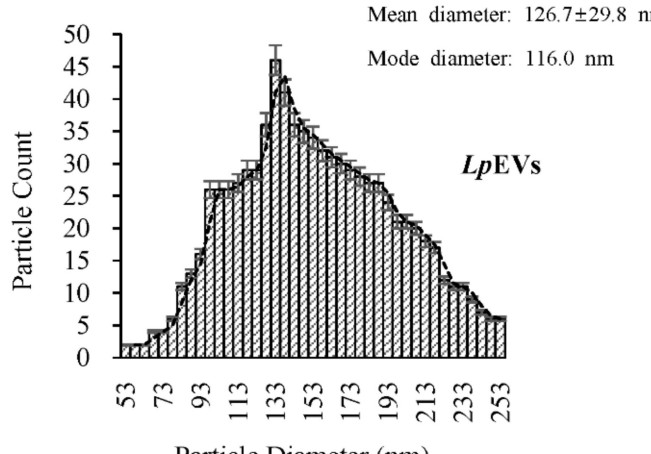

**(C)**

| Sample | Protein Concentration (mg/mL) [1] | Particle Concentration | |
|---|---|---|---|
| | | Per mL [2] | Per mg Protein |
| *Bl*EVs | 21.1 | $2.2 \times 10^{11}$ | $1.04 \times 10^{10}$ |
| *Ba*EVs | 17.0 | $1.9 \times 10^{11}$ | $1.12 \times 10^{10}$ |
| *La*EVs | 23.0 | $3.8 \times 10^{11}$ | $1.65 \times 10^{10}$ |
| hs−*Lh*EVs | 19.7 | $2.03 \times 10^{11}$ | $1.03 \times 10^{10}$ |
| *Lp*EVs | 26.4 | $4.6 \times 10^{11}$ | $1.74 \times 10^{10}$ |

[1] Bradford protein assay.
[2] Nanoparticle tracking analysis (NTA).

**Figure 2.** *Cont.*

**(D)**

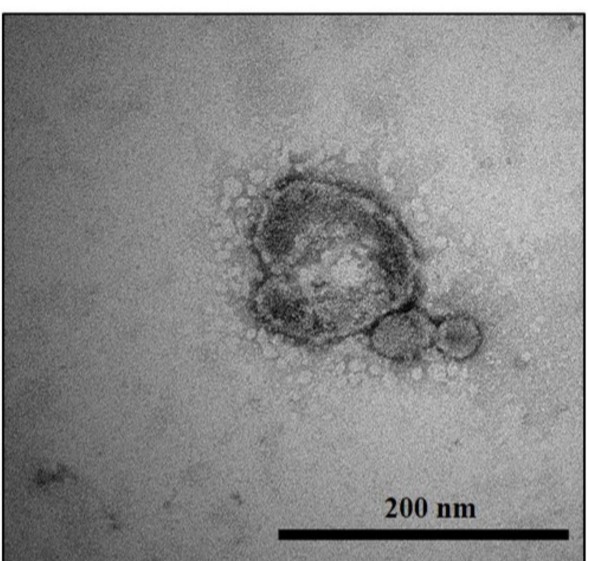
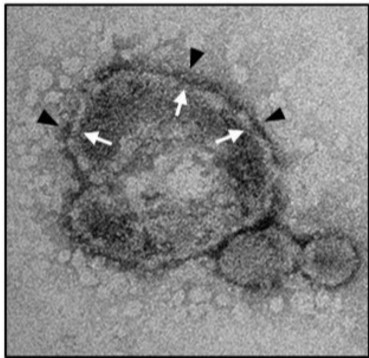

**Figure 2.** Purification and characteristics of probiotic bacteria-derived extracellular vesicles (PB-EVs). (**A**) A representative frame from one of the PB-EV nanoparticle tracking analysis videos is shown. The purified EVs were diluted to 1:1000 in DPBS. (**B**) Particle size and number of PB-EVs determined by nanoparticle tracking analysis (NTA). (**C**) Protein and particle concentration of PB-EVs. (**D**) Cryo-TEM image analysis with the negative staining of *Leuconostoc holzapfelii*-derived extracellular vesicles (hs-*Lh*EVs). The outlined EV images are enlarged, and the lipid bilayer is indicated by black and white arrows, which refer to the potential double membrane of the vesicles. Scale bars, 200 nm. *Bl*EVs, *Bifidobacterium longum*-derived extracellular vesicles; *Ba*EVs, *Bifidobacterium animalis*-derived extracellular vesicles; *La*EVs, *Lactobacillus acidophilus*-derived extracellular vesicles; hs-*Lh*EVs, extracellular vesicles derived from *Leuconostoc holzapfelii* isolated from the human scalp; *Lp*EVs, *Lactobacillus plantarum*-derived extracellular vesicles. All data are reported as the mean ± SD of three independent experiments.

### *3.3. Probiotic Bacteria-EVs Induce HFDP Cell Migration and Proliferations*

To understand whether EVs isolated from PB belonging to the Lactobacillus and Bifidobacterium genera could influence HFDP cell migration, a scratch wound healing assay was performed in the presence of 10 μg/mL of PB-EVs. We found that cell migration was generally increased by 24 h of treatment with all PB-EVs. However, cell migration following treatment with hs-LhEVs significantly increased compared to that with other PB-EVs (Figure 3A,B). As it was revealed that cell migration was enhanced by hs-LhEVs compared to other PB-EVs, we further examined the effects of hs-LhEVs on HFDP cells in dose- and time-dependent manners. With variations according to the treatment concentration and incubation time, the presence of hs-LhEVs was associated with the more rapid healing of the initial wound area than the PBS control. These results demonstrated that HFDP cell exposure to hs-LhEVs accelerated cell migration (Figure 4A,B). We also validated that the PB-EVs exerted a proliferative effect on HFDP cells. We treated the cells with 10 μg/mL of PB-EVs and measured proliferation with the CCK-8 assay after 24 h. HFDP cell proliferation increased from about 7 to 11% in all groups treated with PB-EVs except for hs-LhEVs, where cell proliferation significantly increased by 24% (Figure 3C). Further experiments on the effect of hs-LhEV treatment on cell proliferation revealed a dose-dependent effect. Taken together, these results suggest that PB-EV treatment induce cell migration and increase cell proliferation in HFDP cells, as well as that LhEVs have a greater effect than the other PB-EVs used in this study.

(A)

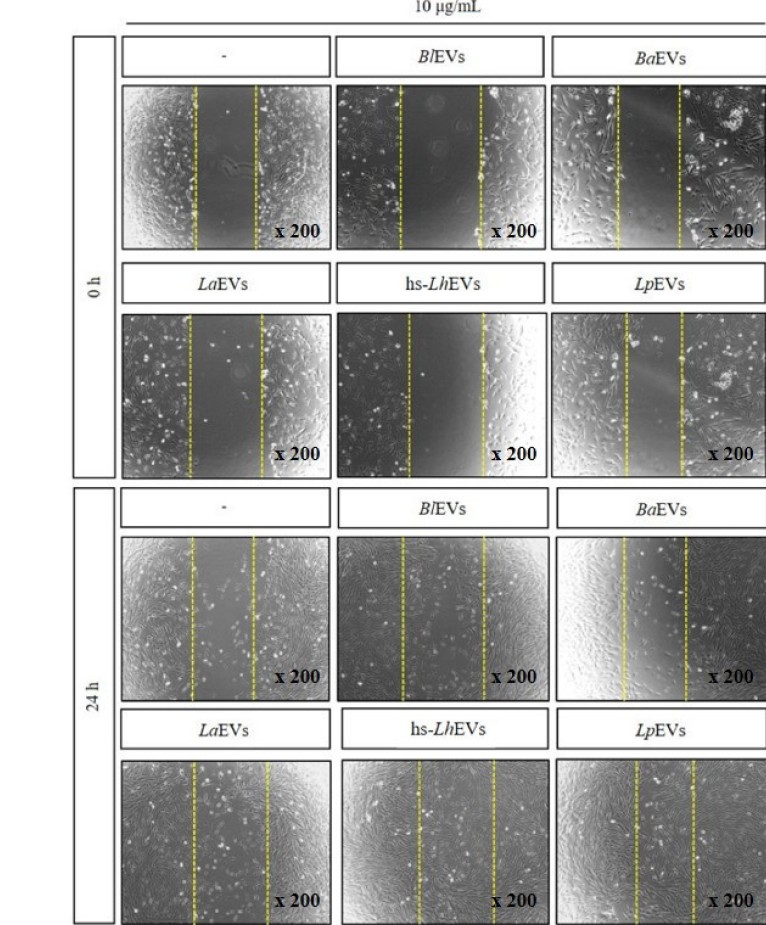

(B)

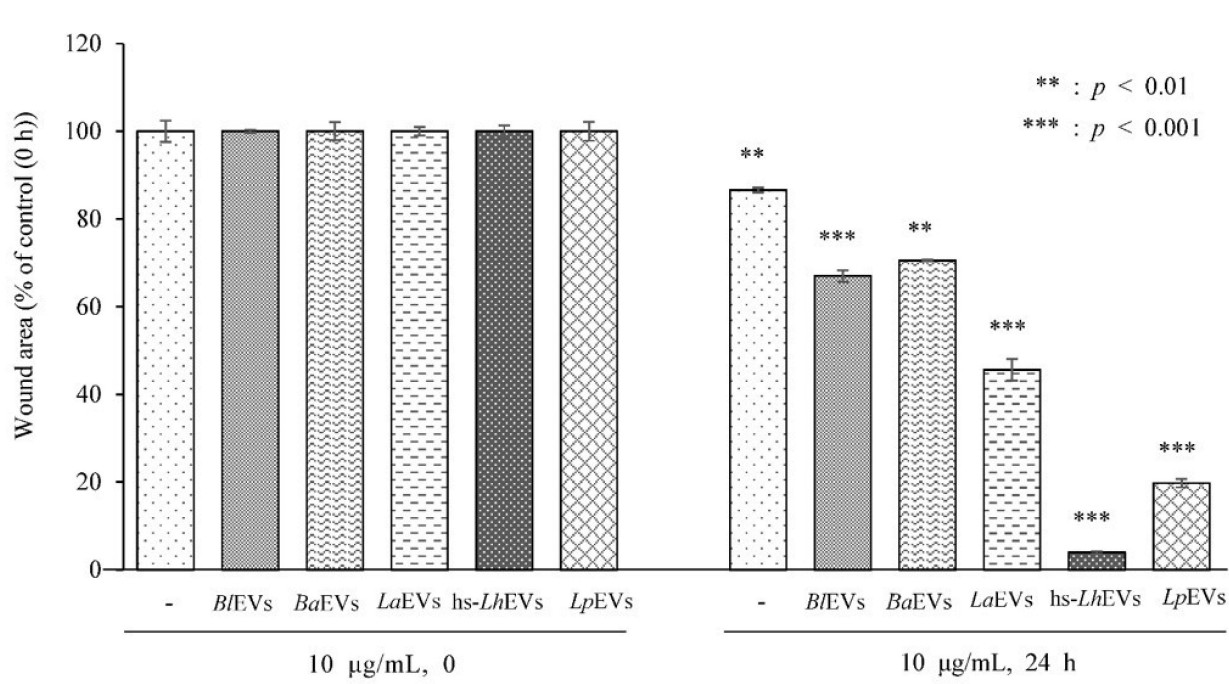

**Figure 3.** *Cont.*

(C)

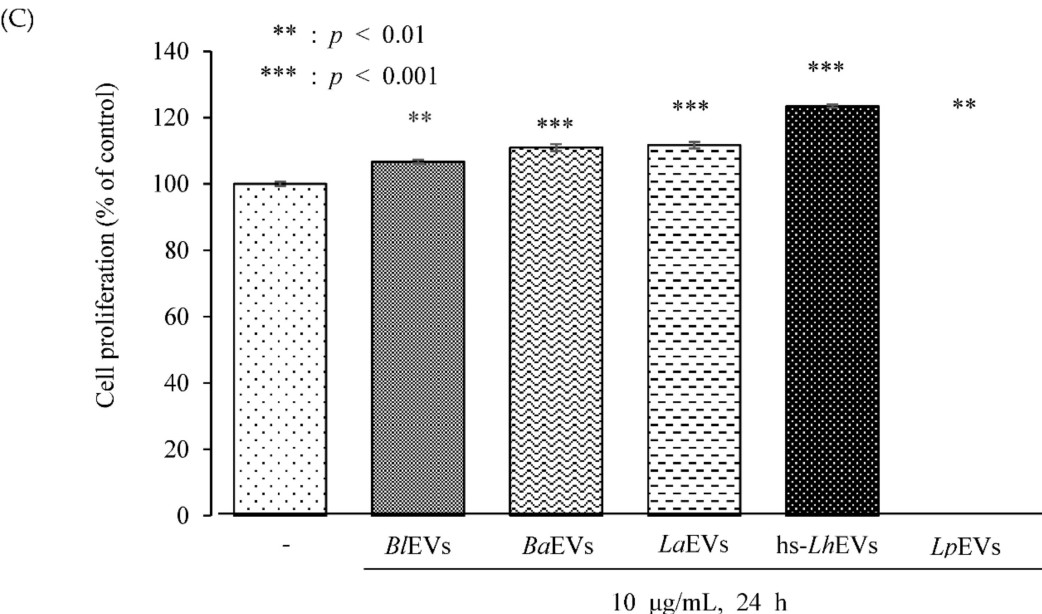

**Figure 3.** Cell migration and proliferation inductive effects of PB-EVs in human follicle dermal papilla cells (HFDPCs). (**A**) Cell migration as determined by a wound healing assay, treated with 10 μg/mL of *Bl*EVs, *Ba*EVs, *La*EVs, *Lh*EVs, or *Lp*EVs for 24 h. (**B**) Quantification of migrated cells shown in (**A**). (**C**) Cell proliferation as assessed with CCK-8-based assay performed on HFDPCs treated with 10 μg/mL of PB-EVs. Mean ± SD of experiments are shown. ** $p < 0.01$; *** $p < 0.001$—Student's *t*-test was used. *Bl*EVs, *Bifidobacterium longum*-derived extracellular vesicles; *Ba*EVs, *Bifidobacterium animalis*-derived extracellular vesicles; *La*EVs; *Lactobacillus acidophilus*-derived extracellular vesicles; hs-*Lh*EVs, extracellular vesicles derived from *Leuconostoc holzapfelii* isolated from the human scalp; *Lp*EVs, *Lactobacillus plantarum*-derived extracellular vesicles.

(**A**)

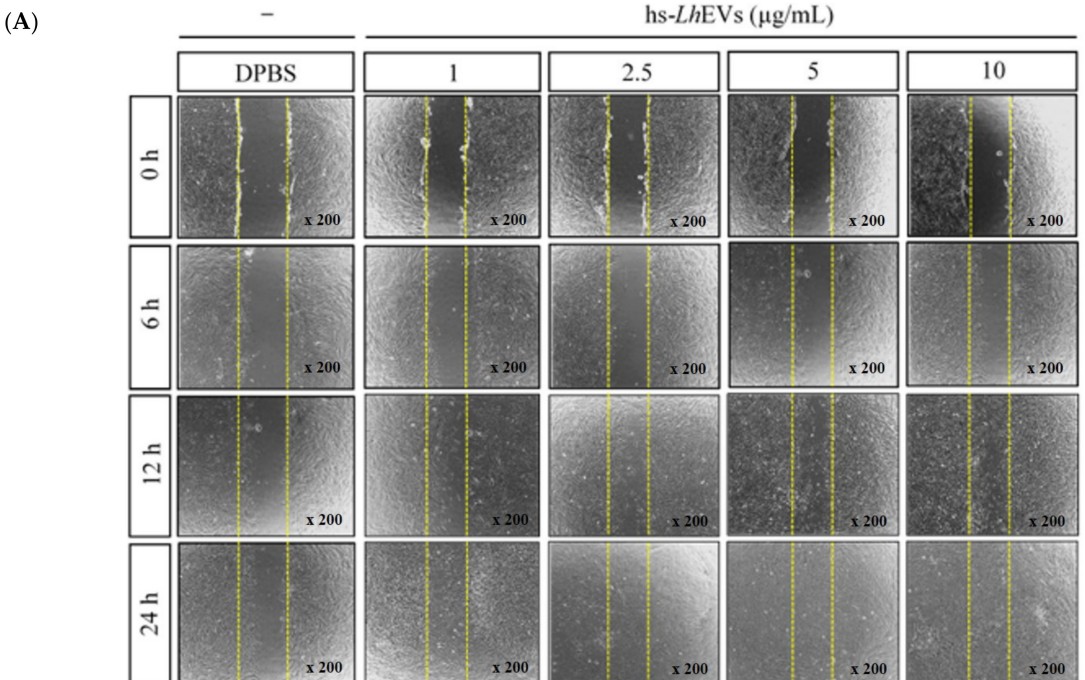

**Figure 4.** *Cont.*

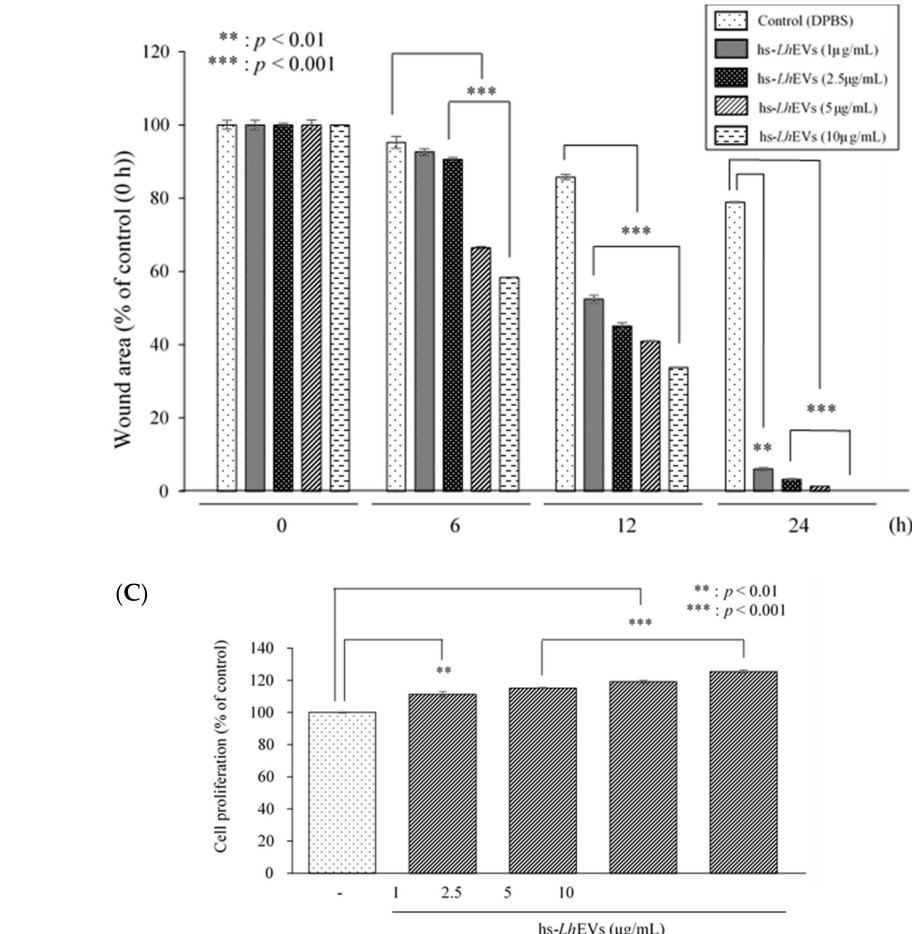

**Figure 4.** Cell migration and proliferation inductive effects of hs-*Lh*EVs in human follicle dermal papilla cells (HFDPCs). (**A**) Cell migration as determined by a wound healing assay, treated with 1, 2.5, 5, 10 μg/mL hs-*Lh*EVs in a time-dependent manner. (**B**) Quantification of migrated cells shown in (**A**). (**C**) Cell proliferation as assessed with a CCK-8-based assay performed on HFDPCs treated with hs-*Lh*EVs in time- and dose-dependent manners. Mean ± SD of experiments are shown. ** *p* < 0.01; *** *p* < 0.001—Student's *t*-test was used. *Bl*EVs, *Bifidobacterium longum*-derived extracellular vesicles.

*3.4. hs-LhEVs Inhibit Apoptosis and Induce Division via Control Cell Cycle in HFDP Cells*

To confirm the change the aforementioned factors, we first investigated how hs-LhEVs affected the cell cycle in vitro. HFDP cells treated with hs-LhEVs at all doses (1, 2.5, 5, and 10 μg/mL) and times (6, 12, and 24 h) affected the sub-G1, G1, S, and G2/M phases of the cell cycle. At 6 h post-treatment, hs-LhEV exposure decreased the sub-G1 phase (apoptotic cells) population in a dose-dependent manner compared to the control at 88.3 ± 3.4%. The sub-G1 phase population was slightly reduced to 87.0 ± 4.1% at 1 μg/mL, 85.7 ± 3.4% at 2.5 μg/mL, and 83.7 ± 4.2% at 5 μg/mL, and it significantly decreased to 67.2 ± 2.1% at 10 μg/mL. In the G1 phase where cell apoptosis appeared from 1 to 10 μg/mL of hs-LhEVs, the largest reduction rate was shown at 10 μg/mL. Additionally, depending on the incubation time, the changing reduction patterns of the sub-G1 phase in the cell cycle were 23.9% at 6 h, 48.6% at 12 h, and 32.3% at 24 h (Figure 5A–C). Conversely, the S and G2/M phase populations (divisional cells) dose-dependently increased following hs-LhEV treatment compared to the control; similarly to sub-G1, these populations showed the biggest differences in 10 μg/mL of hs-LhEVs, and changing the ratio in the S phase appeared to increase the populations by 3 fold at 6 h, 1.5 fold at 12 h, and 1.8 fold at 24 h. Additionally, the G2/M phase population increased by 4 fold at 6 h, 1.7 fold at 12 h, and

1.3 fold at 24 h (Figure 5D). These results indicate that hs-LhEVs regulate apoptosis and division through sub-G1 phase decreases and G2/M phase increases in HFDP cells.

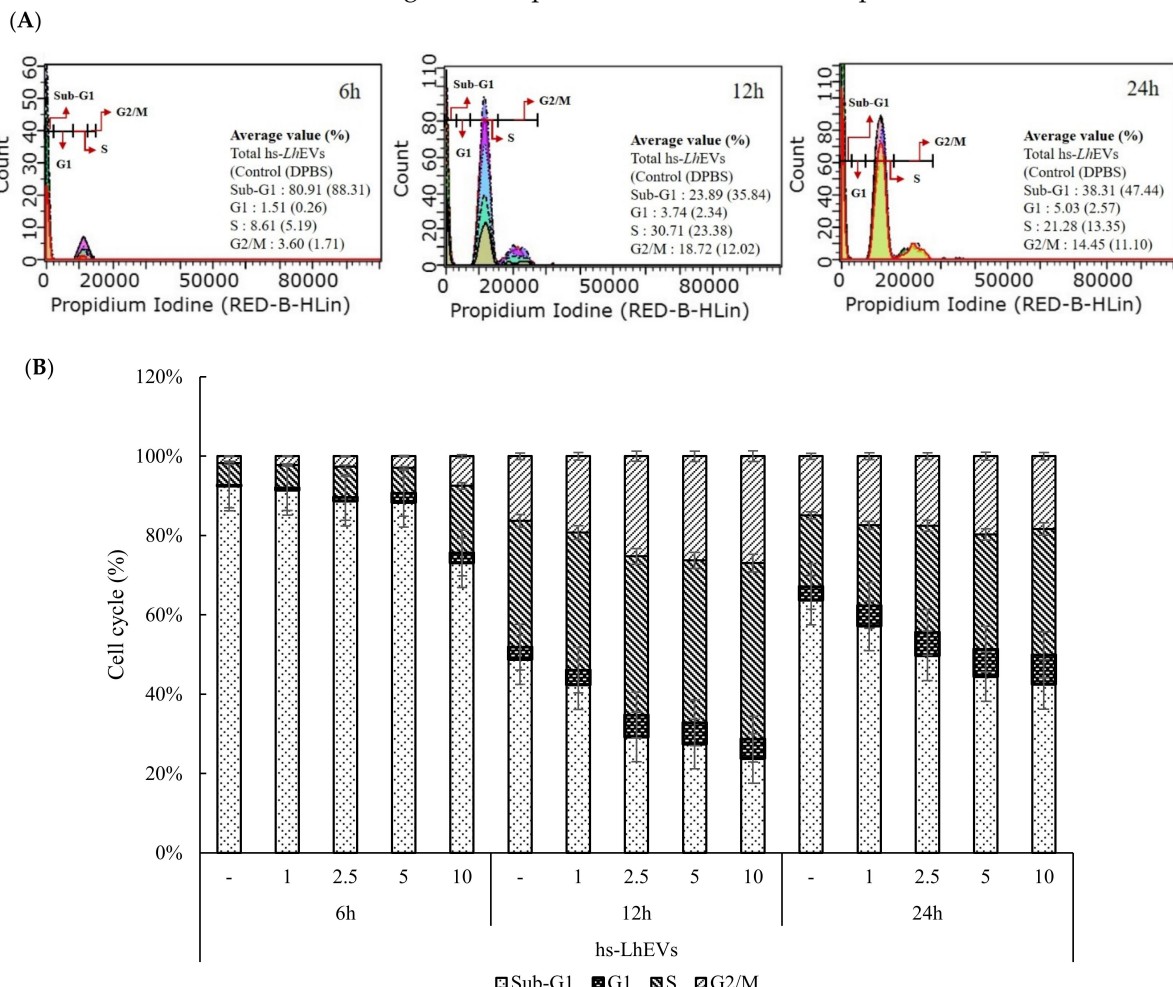

**(C)**

| Sample | Time | Concentration (μg/mL) | SubG1 | G1 | S | G2/M |
|---|---|---|---|---|---|---|
| hs–*Lh*EVs | 6 h | – | 88.31 ± 2.4 | 0.26 ± 0.0 | 5.19 ± 1.6 | 1.71 ± 0.3 |
| | | 1 | 86.96 ± 4.2 | 0.62 ± 0.1 | 5.43 ± 1.8 | 2.12 ± 0.6 |
| | | 2.5 | 85.71 ± 3.2 | 0.93 ± 0.3 | 7.43 ± 2.1 | 2.60 ± 0.5 |
| | | 5 | 83.72 ± 3.8 | 2.17 ± 0.9 | 6.05 ± 1.7 | 2.79 ± 0.4 |
| | | 10 | 67.24 ± 2.9 | 2.30 ± 1.1 | 15.52 ± 2.2 | 6.90 ± 0.7 |
| | 12 h | – | 35.84 ± 4.6 | 2.34 ± 0.8 | 23.38 ± 2.3 | 12.02 ± 1.4 |
| | | 1 | 35.48 ± 3.9 | 3.10 ± 1.2 | 28.99 ± 2.6 | 16.10 ± 2.1 |
| | | 2.5 | 21.60 ± 2.3 | 4.10 ± 0.9 | 29.59 ± 3.1 | 18.68 ± 2.5 |
| | | 5 | 20.06 ± 2.4 | 3.95 ± 1.1 | 29.90 ± 3.4 | 19.21 ± 2.6 |
| | | 10 | 18.42 ± 1.9 | 3.79 ± 0.7 | 34.34 ± 4.1 | 20.87 ± 2.2 |
| | 24 h | – | 47.44 ± 3.3 | 2.57 ± 0.6 | 13.35 ± 1.8 | 11.10 ± 1.4 |
| | | 1 | 44.27 ± 3.1 | 3.98 ± 0.8 | 15.62 ± 2.4 | 13.49 ± 2.2 |
| | | 2.5 | 40.61 ± 2.8 | 4.85 ± 1.1 | 21.96 ± 3.2 | 14.32 ± 1.9 |
| | | 5 | 36.25 ± 2.6 | 5.66 ± 1.4 | 23.56 ± 3.7 | 16.10 ± 2.7 |
| | | 10 | 32.12 ± 2.4 | 5.62 ± 1.6 | 23.98 ± 3.1 | 13.89 ± 1.9 |

**Figure 5.** *Cont.*

(**D**)

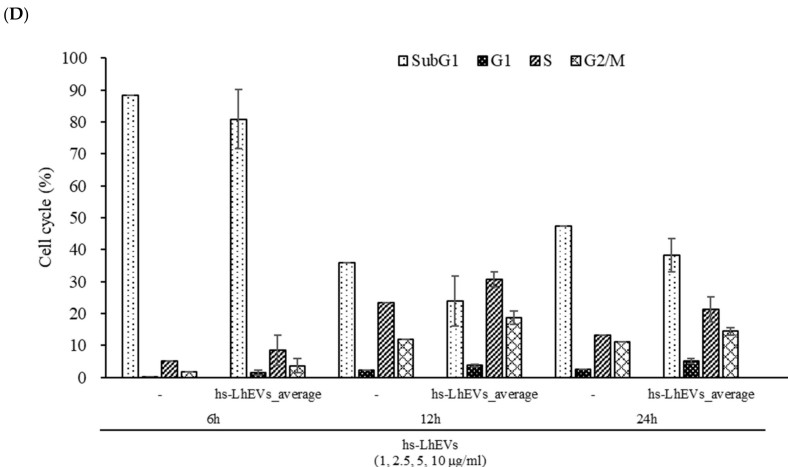

**Figure 5.** Analysis of the cell cycle status of hs-*Lh*EVs discussed in (**A**–**D**). HFDP cells were seeded in a 6-well plate at $3 \times 10^5$ cells in 2 mL/well and cultured with complete medium for 24 h. Cells were starved in serum-free medium for 16 h and treated with 1, 2.5, 5, and 10 μg/mL of hs-*Lh*EVs for 6, 12, and 24 h in same conditioned medium. (**A**) Cell cycle histogram with propidium iodide (P.I) fluorescence profiles treated with hs-*Lh*EVs in dose- and time-dependent manners. (**B**–**D**) Analysis data of cell cycle shown in (**A**). Data are reported as mean ± SD of three independent experiments.

*3.5. hs-LhEV Treatment Regulates Apoptosis-Associated mRNA Expression in Cisplatin-Induced HFDP Cells*

Based on the previous results indicating effects on the cell cycle (Figure 5), we examined whether hs-LhEVs could regulate apoptosis-associated mRNA expression in cisplatin-induced HFDP cells in comparison with the supernatant of hs-LhCM. Reverse transcription PCR (RT-PCR) studies following hs-LhEV treatment showed the expression of anti-apoptotic Bcl-2 and pro-apoptotic Bax, as well as the activation of caspase-3 mRNA, in a dose-dependent manner. These results indicate that hs-LhEV treatment induced Bcl-2 mRNA and Bax expression and reduced caspase-3 activity compared to the control (Figure 6). In the case of hs-LhCM treated at the same concentrations (maximum concentration: 10 μg/mL) as hs-LhEVs, the anti-apoptotic effects in cisplatin-induced HFDP cells were minor compared to hs-LhEVs. These results support the idea that hs-LhEVs remarkably inhibit the effects of cisplatin on the apoptosis-regulatory genes Bcl-2 and Bax in a dose-dependent manner. In addition, hs-LhEVs inhibited caspase-3 mRNA expression and decreased the ratio of Bcl-2 to Bax (Figure 6). Taken together, these results suggest that hs-LhEVs suppress apoptosis-related factors more than CM of hs-Lh and correspond with the cell cycle results in cisplatin-induced HFDP cells (Figure 5).

(**A**)

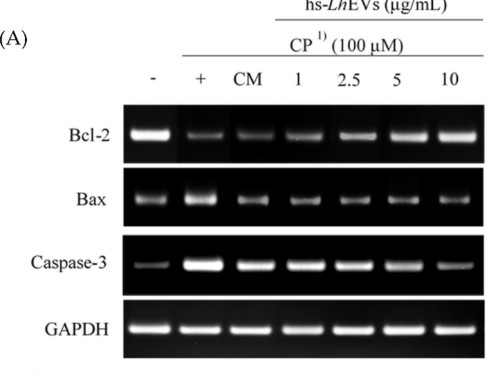

[1]) CP : Cis-diamminedichloroplatinum II (cisplatin)

**Figure 6.** *Cont.*

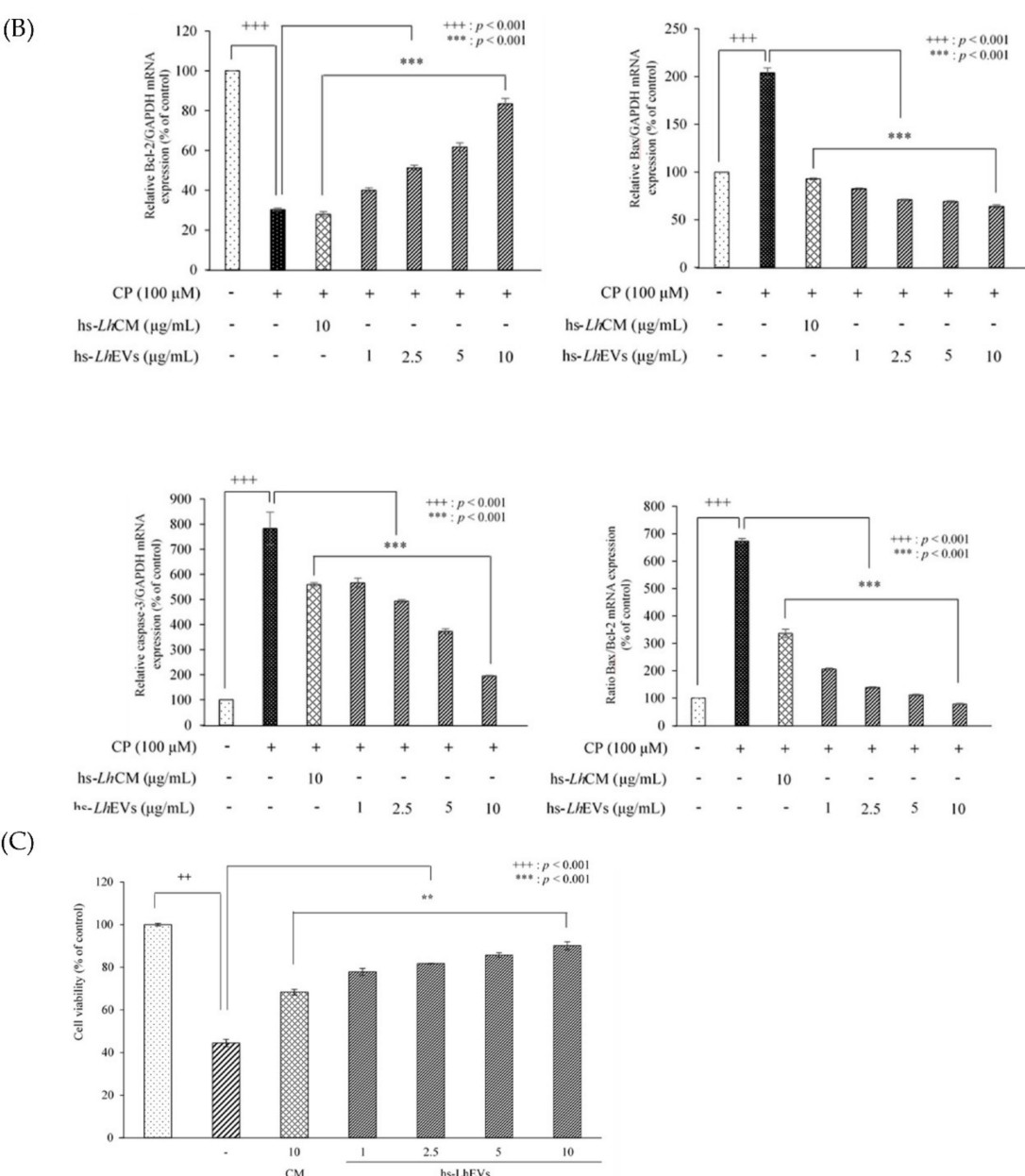

**Figure 6.** Evaluation of the effects of hs-*Lh*EVs on the *Bcl-2*, *Bax*, and *caspase-3* mRNA expression levels using RT-PCR in cisplatin-induced HFDPCs. (**A**) The *Bcl-2*, *Bax*, and *caspase-3* expression levels in HFDPCs after 24 h of treatment. (**B**) Quantification of mRNA expression shown in (**A**). (**C**) Cell viability of hs-*Lh*EVs in cisplatin-induced HFDP cells. Data are reported as mean ± SD of three independent experiments. Statistical analysis was performed with Student's *t*-test at significance levels of ++ $p < 0.01$ (compared to control); ** $p < 0.01$ (compared to hs-LhEVs non-treatment); +++ $p < 0.001$ (compared to control); *** $p < 0.001$ (compared to hs-*Lh*EVs non-treatment).

### 3.6. hs-LhEV Treatment Induces Hair-Inductive-Associated mRNA Activity in HFDP Cells

Next, to validate whether hs-LhEVs affect the hair-inductive activity of HFDP cells, we measured the expression levels of genes associated with Wnt/β-catenin signaling transduction. In detail, we investigated changes in the patterns of hair-related factors in the absence or presence of hs-LhEVs. The mRNA expression of Wnt5A, Wnt10B, and β-catenin in hs-LhEV-treated HFDP cells was significantly increased compared to the PBS control at 24 h (Figure 7). In addition, the expression levels of VSC, Lef1, BAMBI, and BMP-2 were increased in a dose-dependent manner following hs-LhEV exposure (Figure 7). Our

results especially show that treatment with 10 µg/mL of hs-LhEVs remarkably accelerated the expression of Wnt-signaling-related genes. Taken together, these results suggest that hs-LhEVs could promote the hair-inductive activity of HFDP cells more than the PBS or CM controls through the Wnt/β-catenin signaling pathway.

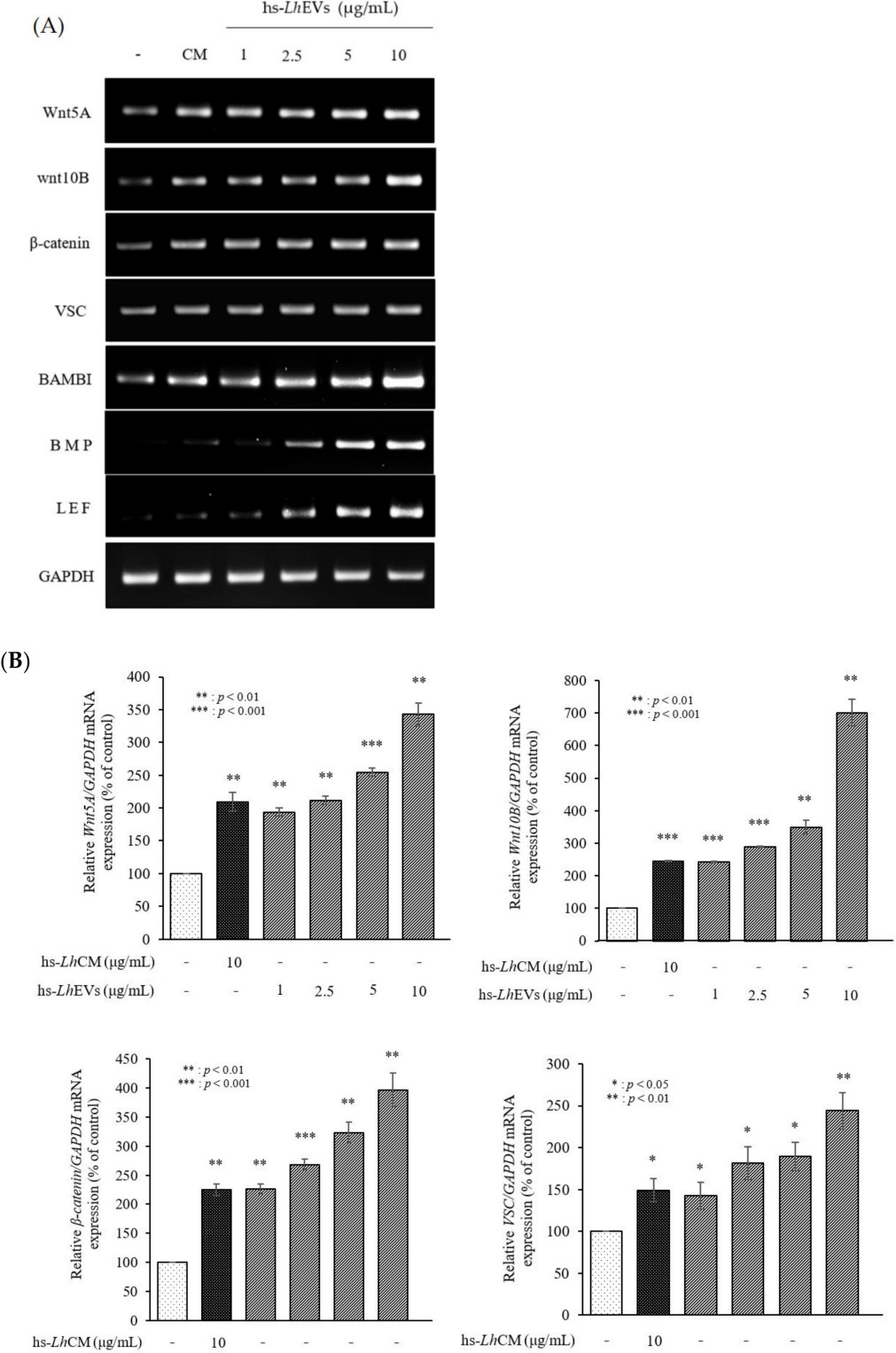

**Figure 7.** *Cont.*

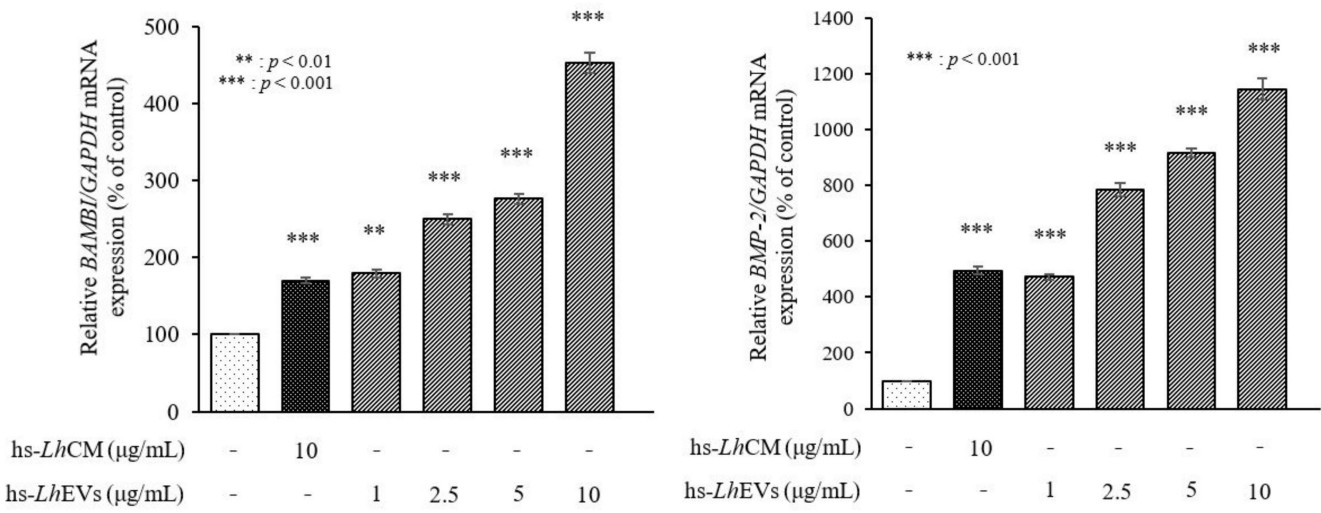

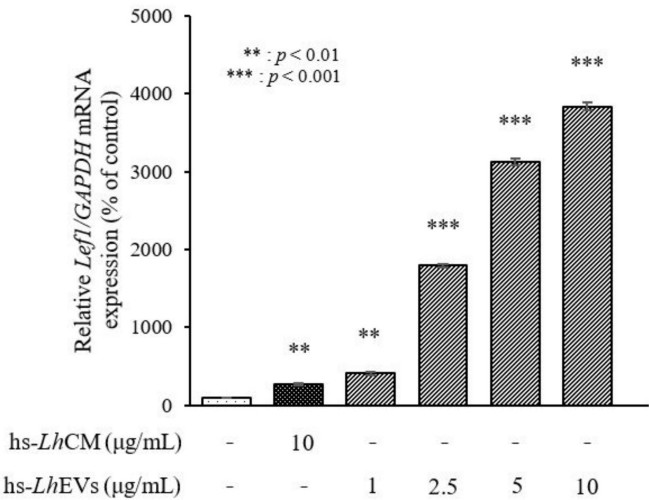

**Figure 7.** Evaluation of the effects of hs-*Lh*EVs on *Wnt5A*, *Wnt10B*, *β-catenin*, versican (*VCAN*), *BAMBI*, *BMP-2*, and *Lef1* mRNA expression levels using RT-PCR in HFDPCs. (**A**) The Wnt5A, Wnt10B, β-catenin, BAMBI, Lef1, BMP-2, and versican (VCAN) expression levels in HFDPCs after 24 h of treatment. (**B**) Quantification of mRNA expression shown in (**A**). Data are reported as mean ± SD of three independent experiments. * $p < 0.05$; ** $p < 0.01$; *** $p < 0.001$—Student's *t*-test was used.

## 4. Discussion

In the modern cultures that values appearance, the hair growth industry is in the spotlight. Most related researchers are conducting studies to find novel materials that promote hair growth in in vivo animal and ex vivo hair follicle culture models at the cellular and molecular levels. In hair growth studies, HFDP cells have been reported to be an efficient tool for screening candidates with hair-inductive activities [18]. In 2013, the International Scientific Association for Probiotics and Prebiotics (ISAPP) reported that probiotics and prebiotics can benefit human nutrition and health, an assertion that is still maintained [54]. *Leuconostoc lactis*, isolated from marine finfish, inhibits the growth of pathogens [55]; *L. mesenteroides* improves immunological function [56]; and *L. pseudomesenteroides* has strong antibacterial properties [57]. Although it has been reported that the Leuconostoc genus has beneficial effects, those of *Leuconostoc holzapfelii* (Lh), particularly the extracellular vesicles derived from human scalp isolates of *L. holzapfelii*, are completely unknown. Here, we report for the first time that EVs derived from a human scalp isolate of *L. holzapfelii* can induce hair growth. The characterization of PB-EVs obtained from conditioned medium supernatant revealed round vesicles with a mean diameter between 100 and

130 nm. Interestingly, we found that the different PB-EV types had an average diameter of 120 nm, and hs-LhEVs derived from human-scalp-isolated *L. holzapfelii* had a smaller diameter of 104.6 nm (Figure 2B). Apoptosis, a process by which cell death is genetically regulated, plays a pivotal role in maintaining the homeostasis of an organism when cells are exposed to pathogenic stimuli or cellular toxicity [58]. Apoptosis effects are regulated by a complex molecular signaling pathway with the key regulatory components of B-cell CLL/lymphoma 2 (Bcl-2), Bcl-2-associated × protein (Bax), and caspases [19,59,60]. Treatment with cisplatin (CP) results in apoptosis from mitochondrial death [21,22] and ROS generation in HFDPCs and keratinocytes [61]. A previous report indicated that treatment with macrophage-derived extracellular vesicles (MAC-EVs) promotes the migration and proliferation of DP cells, and cell proliferation, apoptosis, and various factors play crucial roles in the biology of hair growth [62]. These results suggest that hs-Lh might produce EVs with a higher protein content than other PBs (Figure 2C). In our wound healing and CCK-8 colorimetric assays, hs-LhEV treatment significantly increased cell migration and proliferation by 84% and 24%, respectively, compared to other PB-EVs (Figures 3 and 4). The use of lactic acid bacteria lysates has shown that cell proliferation and wound healing are promoted in keratinocytes [63]. Apoptosis, cell proliferation, and hair-inductive genes comprise a large proportion of the biology of hair growth. Apoptosis is affected by apoptosis-related factors such as caspase-3, Bcl-2, and Bax, and these BAX/BAK signal transducers activate the effector caspases of caspase-3 and caspase-7 [64,65]. In this study, we showed that hs-LhEVs could regulate apoptosis-regulated factors, such as caspase-3, Bcl-2, and Bax, more than hs-LhCM and could reduce apoptosis through sub-G1 phase inhibition and G2/M phase induction in HFDP cells (Figures 5 and 6). Specifically, we suggest that hs-LhEVs may regulate apoptosis and division through cell cycle progression. Moreover, cell proliferation and hair growth development have been known to signal transduction through the Wnt pathway, which is involved in the developmental regulation of cell processes such as cell proliferation, survival, migration, and polarity [66]. Factors related to homeostasis, embryonic development, proliferation, and differentiation on cells confirmed in this study are: Wnt5A; Wnt10B; β-catenin; Lef1 (lymphoid enhancer-binding factor 1), a type of hair matrix gene; BAMBI (BMP and active membrane-binding inhibitor homolog), a component of Wnt signal transduction; BMP-2 (bone morphogenetic protein 2), hair-inductive activity; and VSC (versican), which is associated with growth phase induction [67]. Treatment with hs-LhEVs could promote hair-inductive activities via the regulation of the Wnt signaling pathway containing Wnt5A, Wnt10B, β-catenin, VSC, BAMBI, BMP-2, and Lef1 (Figure 7). In summary, we have demonstrated that *Leuconostoc holzapfelii*, which exists in the human scalp, produces EVs with different characteristics from other general probiotic bacteria that may be able to regulate apoptotic factors such as Bcl-2, Bax, and caspase-3. Additionally, hs-LhEVs appear to stimulate cell division, migration, and proliferation, and they present hair-inductive activity through the Wnt/β-catenin signaling pathway in vitro (Figure 8). In conclusion, there is potential for the future development of hs-LhEVs in cosmetics and pharmaceuticals that promote hair growth. In addition, we confirmed that hs-LhEVs increase the number of hairs compared to placebo through a clinical trials (Supplementary Materials Figures S1–S3). For further study, we are going to conduct more in-depth research regarding hs-LhEVs through organ culture methods.

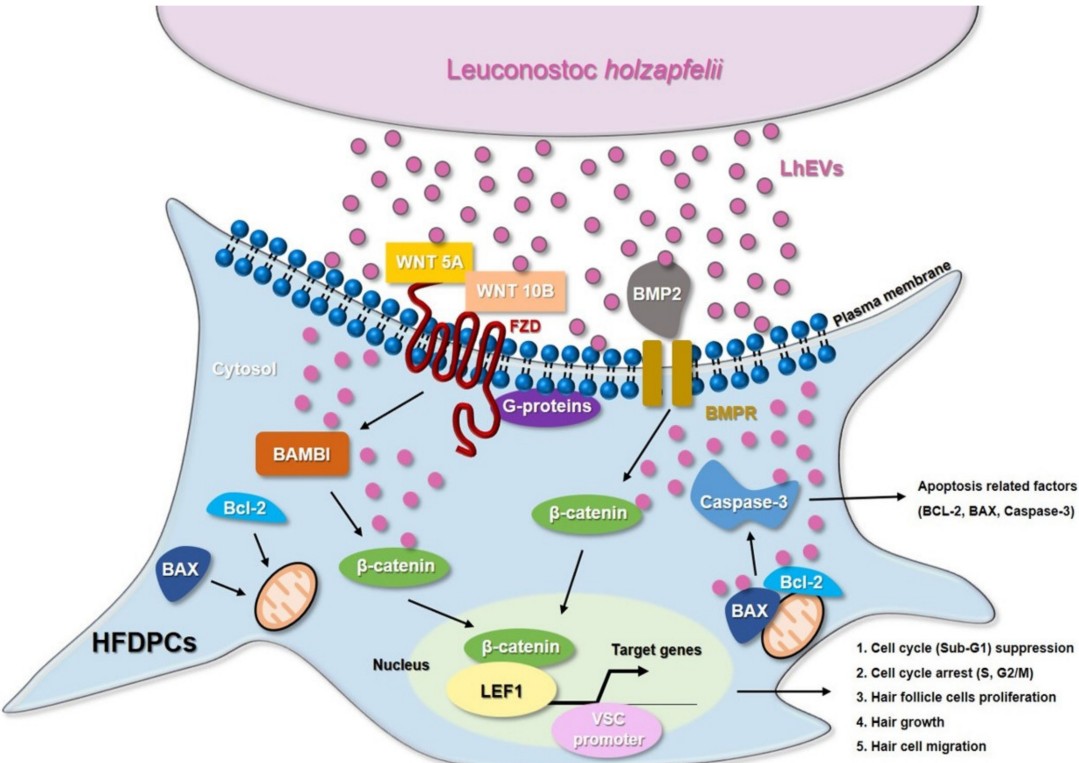

**Figure 8.** Schematic of the apoptosis inhibition and hair growth effects of extracellular vesicles derived from *Leuconostoc holzapfelii* isolated from human scalp tissue (hs-*Lh*EVs).

**Supplementary Materials:** The following supporting information can be downloaded at https://www.mdpi.com/article/10.3390/cimb44020058/s1. Figure S1: Age distribution of subjects in the clinical trials related to hair growth; Figure S2: Increase in the number of hairs in hs-LhEVs by clinical trial period; Figure S3: Changes in average hair count (N/cm2) compared to placebo.

**Author Contributions:** Data curation, J.W.M.; formal analysis, Y.C.Y., B.H.A. and S.H.P.; funding acquisition, H.C.K.; methodology, Y.C.Y. and J.W.M.; project administration, H.C.K.; resources, B.H.A. and H.C.K.; supervision, H.C.K.; writing—original draft, Y.C.Y.; writing—review and editing, K.R.L., H.C.K. and S.H.P. All authors have read and agreed to the published version of the manuscript.

**Funding:** This research received no external funding.

**Institutional Review Board Statement:** Not applicable.

**Informed Consent Statement:** Not applicable.

**Data Availability Statement:** Not applicable.

**Conflicts of Interest:** The authors declare no conflict of interest.

## Abbreviations

| | |
|---|---|
| EVs | Extracellular vesicles |
| PB-EVs | Probiotic bacteria derived-extracellular vesicles |
| hs-Lh | Human scalp-derived *Leuconostoc holzapfelii* |
| hs-LhCM | Conditioned medium of *Leuconostoc holzapfelii* in human scalp |
| hs-LhEVs | Extracellular vesicles derived from *Leuconostoc holzapfelii* in human scalp |
| BAMBI | Activin membrane-bound inhibitor homolog |
| BMP2 | Bone morphogenetic protein 2 |
| Lef1 | Lymphoid enhancer-binding factor |
| VSC | Versican |

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
