# Peer review of "Stimulatory Effects of Extracellular Vesicles Derived from Leuconostoc holzapfelii That Exists in Human Scalp on Hair Growth in Human Follicle Dermal Papilla Cells"

_cimb, doi:10.3390/cimb44020058_

Round 1
Reviewer 1 Report
An original study investigating the effects of extracellular vesicles derived from leuconostoc holzapfelii on hair growth in human hair follicle dermal papilla cells, showing that extracellular vesicles derived from leuconostoc holzapfelii increase cell proliferation, migration, and regulate the cell cycle, also modulating the mRNA expression of hair growth related genes in vitro.
Only minor queries:
Page 2 line 47 you should add: "Also blue lights treatments have been proposed." and cite an article such as doi: 10.1007/s10103-021-03327-9.
Please specify the SPSS version you used in the statistical analysis subsection in order to calculate significance.
Author Response
Thank you very much for your interest in our article.
It has been applied the parts reviewer requested, and the related details are as follows. Also, upload the file with the correction applied.
-------------------------------------------------------------------------------------
(Q.1) Page 2 line 47 you should add: "Also blue lights treatments have been proposed." and cite an article such as doi: 10.1007/s10103-021-03327-9.
(A.1) We searched for related papers and applied the contents and papers. Modifications were indicated in red text on article.
[Modificated contents] As well as, blue lights treatments have been proposed as additional methods of hair loss treatment [13].
(Q.2) Please specify the SPSS version you used in the statistical analysis subsection in order to calculate significance.
(A.2) We searched for related papers and applied the contents and papers. Modifications were indicated in red text on article. We have marked the SPSS version.
[Modificated contents] Statistical analyses were conducted with the non-parametric Kruskal–Wallis and Mann–Whitney analysis using SPSS version 20.0 program (SPSS Inc., Armonk, NY).

Reviewer 2 Report
The paper of Yeo Cho Yoon et al investigated the role of Leuconostoc holzapfelii vesicels on the hair growth on Human hair follicle dermal papilla cells. The paper is interesting for a broad public. However, description of methods should be improved and the results/discussion section rewritten.
Introduction;
Line 43/44: pls mention also the duration and the percentage of the hair of each phase of hair growth, since this is relevant for different pathologies.
Line 45: “surgical operation”please specify
The Wnt/β-catenin signaling pathway could be better explained: including the mechanism that plays a preeminent role in maintaining cellular homeostasis, embryo development, cell proliferation and differentiation, apoptosis, and inflammation-associated cancer.
Methods:
Fig 1 is not sharp
The relevance of all the experiments as well as the precise methods including quality control are not clear in all of the experiments, please improve.
Statistics is not well enough explained. T-test is not mentioned
Results
Should be rewritten. Some of the text rather fit in discussion for example Line 353 355, line 432, 454 and further, 471 and further
Discussion
Is a bit thin compared to the results.
Line 528-535: better belongs to the introduction
“Herein, we report for the first time that EVs derived from a human scalp isolate of L. holzapfelii can induce hair growth” please explain further, according to which experiments?
“Namely, these results prove that hs-LhEVs can regulate apoptosis and division through cell cycle progression” is a very strong expression. Please provide more information about the experiment (how many times repeated, what quality control)
“In conclusion, hs-LhEVs suggest the possibility of development into cosmetics and pharmaceuticals that promote hair growth in the future.”
What about trying in vivo experiments priori to development of cosmetics?
Author Response
First of all, thank you for your interest in our article. We have applied and corrected your valuable comments as much as possible. The corrected part is indicated in red text, and the file is attached and delivered, so please check it. In addition, one additional author has been added to this article. Thank you.

Round 2
Reviewer 2 Report
The paper has significantly improved